# Role of Prednisolone in Platelet Activation by Inhibiting TxA_2_ Generation through the Regulation of cPLA_2_ Phosphorylation

**DOI:** 10.3390/ani13081299

**Published:** 2023-04-11

**Authors:** Sanggu Kim, Preeti Kumari Chaudhary, Soochong Kim

**Affiliations:** Laboratory of Veterinary Pathology and Platelet Signaling, College of Veterinary Medicine, Chungbuk National University, Cheongju 28644, Republic of Korea

**Keywords:** prednisolone, thromboxane generation, 2-MeSADP, ERK, platelet

## Abstract

**Simple Summary:**

Although glucocorticoids are frequently used for a variety of purposes, the effects and mechanisms of glucocorticoids on platelets are not fully understood. The present study investigates the effect of prednisolone and its mechanism of action on the regulation of platelet function. The results demonstrated that prednisolone affects platelet function by inhibiting thromboxane A_2_ (TxA_2_) generation through the regulation of cPLA_2_ phosphorylation, providing knowledge of glucocorticoids in coagulation and bleeding disorders.

**Abstract:**

Glucocorticoids have been commonly used in the treatment of inflammation and immune-mediated diseases in human beings and small animals such as cats and dogs. However, excessive use can lead to Cushing’s syndrome along with several thrombotic and cardiovascular diseases. Although it is well-known that glucocorticoids exert a significant effect on coagulation, the effect of cortisol on platelet function is much less clear. Thus, we aimed to study the effects of prednisolone, one of the commonly used glucocorticoids, on the regulation of platelet function using murine platelets. We first evaluated the concentration-dependent effect of prednisolone on 2-MeSADP-induced platelet function and found that the 2-MeSADP-induced secondary wave of aggregation and dense granule secretion were completely inhibited from 500 nM prednisolone. Since 2-MeSADP-induced secretion and the resultant secondary wave of aggregation are mediated by TxA_2_ generation, this result suggested a role of prednisolone in platelet TxA_2_ generation. Consistently, prednisolone did not affect the 2-MeSADP-induced aggregation in aspirinated platelets, where the secondary wave of aggregation and secretion were blocked by eliminating the contribution of TxA_2_ generation by aspirin. In addition, thrombin-induced platelet aggregation and secretion were inhibited in the presence of prednisolone by inhibiting the positive-feedback effect of TxA_2_ generation on platelet function. Furthermore, prednisolone completely inhibited 2-MeSADP-induced TxA_2_ generation, confirming the role of prednisolone in TxA_2_ generation. Finally, Western blot analysis revealed that prednisolone significantly inhibited 2-MeSADP-induced cytosolic phospholipase A_2_ (cPLA_2_) and ERK phosphorylation in non-aspirinated platelets, while only cPLA_2_ phosphorylation, but not ERK phosphorylation, was significantly inhibited by prednisolone in aspirinated platelets. In conclusion, prednisolone affects platelet function by the inhibition of TxA_2_ generation through the regulation of cPLA_2_ phosphorylation, thereby shedding light on its clinical characterization and treatment efficacy in dogs with hypercortisolism in the future.

## 1. Introduction

Prednisolone is one of the most commonly used glucocorticoids in the treatment of various diseases including inflammatory and immune-mediated diseases. Prednisolone has the ability to reduce the recruitment of inflammatory cells including eosinophils, lymphocytes, mast cells, and dendritic cells by reducing the production of chemotactic mediators and adhesion molecules and the ability of inflammatory cells to survive [1]. Due to the high efficacy of glucocorticoids, the prescription rate has climbed high during the past years, leading to excessive glucocorticoid abuse that raises blood cortisol levels, resulting in iatrogenic Cushing’s syndrome [2,3,4,5].

Cushing’s syndrome is a disorder characterized by polyuria, polydipsia, muscle and skin atrophy, weight loss, truncal obesity, and hair loss [6]. Excessive use of glucocorticoids in thrombotic, hemostatic, and cardiovascular diseases is associated with increased morbidity and mortality [7]. Glucocorticoid medication has been shown to increase the incidence of hypertension, hyperglycemia, and hyperglyceridemia leading to coronary and ischemic heart disease, heart failure, and unexpected death [8,9]. Moreover, high levels of glucocorticoids have been shown to elevate levels of von Willebrand factor (vWF), anti-hemophilic factor, fibrinogen, plasminogen activator inhibitor-1, and platelet count [10,11,12], which can result in embolic disorder and further increase the risk of the coagulative disorder [13]. The pathogenesis and clinical symptoms of Cushing’s syndrome are similar in both humans and dogs. However, the incident rate is 1000 times higher in dogs compared to humans [6]. As a result, Cushing’s syndrome-induced thromboembolic complications are around four times more common in dogs which leads to a four times higher mortality rate, making it a disease of interest in veterinary medicine [14].

It has been well-established that platelets play a crucial role in thrombosis and hemostasis and have been known to play a vital role in the development of blood-related disorders including diabetes and cardiovascular diseases in humans. Various signaling mechanisms are involved in the activation of platelets. Initially, platelets are activated by the exposure of collagen and vWF from the extracellular matrix upon vascular damage. Activated platelets further release adenosine diphosphate (ADP) and generate thromboxane A_2_ (TxA_2_) that enables the recruitment of circulating platelets, activates the recruited platelets, and forms the hemostatic plug in the presence of thrombin by converting fibrinogen to fibrin. Collagen and vWF induce platelet activation by the glycoprotein (GP) VI-mediated signaling pathway through phospholipase C (PLC) γ2 activation [15]. In contrast, G protein-coupled receptor (GPCR) agonists ADP, TxA_2_, and thrombin cause platelet aggregation by the activation of the PLCβ pathway. ADP stimulates G_q_-coupled P2Y_1_ and G_i_-coupled P2Y_12_ receptor-mediated signaling pathways [16]. Likewise, thrombin activates platelets by coupling to G_q_ and G_12/13_ through protease-activated receptors (PARs) [17]. TxA_2_ activates platelets through the thromboxane prostanoid (TP) receptor by coupling to G_q_ and G_12/13_ [18,19]. In addition, ADP-induced TxA_2_ generation is regulated by calcium binding to cytosolic phospholipase A2 (cPLA2) [20,21]. Thrombin-induced TxA_2_ generation is mediated by the P2Y_12_ receptor [22]. The generated TxA_2_ then causes w positive-feedback effect on platelet functional response [23].

Several studies have suggested the occurrence of hyper-coagulation in patients with Cushing’s syndrome whose blood cortisol levels are high [12,24,25]. The study of Manetti et al. has shown that venous thromboembolism occurred in 7.5% of Cushing’s syndrome patients, whereas it did not occur in patients with remission of the disease [26]. Additionally, glucocorticoid administration is regarded as a first-line treatment option in several bleeding disorders, including immune thrombocytopenic purpura and acquired hemophilia A [27]. In contrast, in vitro studies using platelet-rich plasma (PRP) have shown that high glucocorticoid levels exert an inhibitory effect on platelet function [28,29,30]. Until now, the mechanisms involved in the effects of glucocorticoids on the production of thromboxane in platelets have not been elucidated.

In this study, we demonstrated the effect of prednisolone on platelet function and its underlying mechanism using washed murine platelets. We have shown that prednisolone inhibits 2-MeSADP-induced platelet secretion and the resultant secondary wave of aggregation. In aspirinated platelets, prednisolone had no effect on the 2-MeSADP-induced platelet aggregation as well as secretion, suggesting that prednisolone exerts its antiplatelet effect by regulating TxA_2_ generation. We have further shown that 2-MeSADP- and thrombin-induced TxA_2_ generation and 2-MeSADP-induced cPLA_2_ phosphorylation are inhibited in the presence of prednisolone. In conclusion, prednisolone affects platelet function by inhibiting TxA_2_ generation through the regulation of cPLA_2_ phosphorylation.

## 2. Materials and Methods

### 2.1. Materials

2-MeSADP, thrombin, apyrase, prostaglandin E_1_ (PGE_1_), sodium citrate, prednisolone, and acetylsalicylic acid (ASA) were purchased from Sigma (St. Louis, MO, USA). Anti-phospho-cPLA_2_ (Ser505), anti-cPLA_2_, anti-phospho-ERK (Thr202/Tyr204), and anti-total-ERK antibodies were purchased from Cell Signaling Technology (Beverly, MA, USA). HRP-linked secondary antibody was purchased from Santa Cruz Biotechnology (Santa Cruz, CA, USA). Thromboxane B_2_ (TxB_2_) ELISA kit was purchased from Enzo Life Sciences (Exeter, UK). All other reagents were of reagent grade.

### 2.2. Animals

Thirty C57BL/6 mice were purchased from Orient Bio, Inc. at the age of eight weeks (Seongnam-si, Gyeonggi-Do, Republic of Korea) and housed in a semi-pathogen-free facility. No more than 5 mice were housed per cage and mice were freely provided with food and water under constant conditions of temperature (23 °C) and humidity (60%). All animal procedures were approved by the Chungbuk National University Animal Ethics Committee (CBNUA-873-15-02).

### 2.3. Platelet Preparation

Blood was collected via cardiac puncture from healthy mice in 3.8% sodium citrate as described previously [31]. PRP was obtained by centrifugation at 100× *g* for 10 min at room temperature (RT) and 1 µM PGE_1_ was added. For aspirin treatment, the PRP was treated with 1 mM ASA for 30 min at 37 °C. By centrifuging plasma at 400× *g* for 10 min, platelet pellets were obtained from the plasma and re-suspended in Tyrode’s buffer (pH 7.4) with 0.05 units/mL of apyrase. The platelet count was adjusted to 1 × 10^8^ cells/mL.

### 2.4. Platelet Aggregation and Secretion Assay

The light transmission through a 250 μL sample of washed platelets (1 × 10^8^ cells/mL) at 37 °C in a Lumi-aggregometer (Chrono-Log, Havertown, PA, USA) was measured to assess agonist-induced platelet aggregation while stirring at 900 rpm. Amounts of 250 nM, 500 nM, and 1000 nM prednisolone were pre-incubated for 5 min with washed platelets before stimulation with agonists. ATP release from platelets was assessed using a luciferin luciferase reagent to detect platelet dense granule secretion.

### 2.5. Measurement of TxA_2_ Generation

Washed platelets (1 × 10^8^ cells/mL) were stimulated for 3.5 min in a platelet aggregometer under 900 rpm stirring conditions at 37 °C and the reaction was halted by snap-freezing the sample in liquid nitrogen. Samples were stored at −80 °C until TxB_2_ analysis was performed. Briefly, samples were centrifuged at 3000× *g* for 10 min at 4 °C after being thawed at RT, and the supernatant was diluted (1:20) with buffer. The diluted samples were measured to evaluate the levels of TxB2 in duplicates using a TxB_2_ ELISA Kit (Enzo Life Sciences, Exeter, UK), according to the manufacturer’s instructions.

### 2.6. Western Blot Analysis

Washed platelets were activated with 2-MeSADP for 2 min at 37 °C under stirring conditions, and the reaction was stopped by adding 6.6 N perchloric acid. Platelet samples were washed with distilled water, re-suspended with sample buffer, and denatured for 10 min. Platelet samples were separated on a 10% SDS polyacrylamide gel and transferred to polyvinylidene difluoride membranes. Nonspecific binding sites were blocked by SuperBlock^®^ blocking buffer (Thermo Fisher Scientific, Waltham, MA, USA) at RT for 1 h, and membranes were incubated overnight with anti-phospho-cPLA_2_ (Ser505), anti-cPLA_2_, anti-phospho-ERK (Thr202/Tyr204), or anti-total-ERK antibodies diluted to a concentration of 1:1000 with gentle agitation. After 3 washes for 5 min each with Tris-buffered saline with 0.1% Tween 20, the membranes were incubated with appropriate secondary antibody at RT for 1 h. After washing, membranes were incubated with chemiluminescence substrate (Pierce, Rockford, IL, USA) for 5 min at RT, and immune reactivity was detected by using iBright^TM^ CL1500 (Thermo Fisher Scientific, Waltham, MA, USA).

### 2.7. Statistical Analysis

All statistical calculations were performed using Prism software (version 9.1). The data were presented as mean ± standard error (SE). Statistical significance was established using the one-way analysis of variance (ANOVA) followed by a post hoc Dunnett’s multiple comparison test. The normality test was conducted by using a Shapiro–Wilk test and data were found to be normally distributed (*p* > 0.05).

## 3. Results

### 3.1. Prednisolone Regulates 2-MeSADP-Induced Secondary Wave of Platelet Aggregation and Secretion in Platelets

In order to determine the role of prednisolone in platelet function, we first stimulated the platelets with 2-MeSADP and measured the platelet aggregation and dense granule secretion. As shown in Figure 1A, platelet aggregation and dense granule secretion induced by 2-MeSADP were inhibited in the presence of prednisolone in a concentration-dependent manner. 2-MeSADP-induced dense granule secretion and the secondary wave of aggregation were not completely inhibited by 250 nM prednisolone, while both dense granule secretion and the secondary wave of aggregation were completely inhibited in the presence of 500 nM prednisolone. Therefore, we used 500 nM prednisolone throughout the experiments. Interestingly, prednisolone treatment did not affect 2-MeSADP-induced primary aggregation. Thus, our findings suggest that prednisolone inhibits 2-MeSADP-induced dense granule secretion and the resultant secondary wave of aggregation.

### 3.2. The Effect of Prednisolone in 2-MeSADP-Induced Platelet Function in Aspirinated Platelets

Consistent with the previous result, prednisolone inhibited 2-MeSADP-induced platelet secretion and the resultant secondary wave of aggregation at both low and high concentrations of agonist (Figure 2A). It is well-established that ADP-induced platelet secretion and the secondary wave of aggregation require TxA_2_ generation [32,33]. In order to evaluate the effect of prednisolone on TxA_2_ generation, we stimulated the platelets with 2-MeSADP in the presence of aspirin, which blocks the positive-feedback effect of the generated TxA_2_. As shown in Figure 2B, prednisolone showed no further inhibition at both low and high concentrations of 2-MeSADP-induced platelet aggregation in aspirinated platelets compared to non-aspirinated platelets. These results suggest that prednisolone may inhibit 2-MeSADP-induced platelet aggregation and dense granule secretion by inhibiting TxA_2_ generation.

### 3.3. Prednisolone Regulates Only Low Concentrations of Thrombin-Induced Platelet Aggregation and Secretion

To check the role of prednisolone in PAR-mediated platelet function, we stimulated the platelets with various concentrations of thrombin. As shown in Figure 3A, prednisolone inhibited a low concentration of thrombin-induced platelet aggregation and dense granule secretion. However, prednisolone could only partially inhibit platelet secretion when stimulated with a high concentration of thrombin. Additionally, prednisolone did not show any further inhibitory effect on thrombin-induced platelet aggregation and dense granule secretion in aspirinated platelets (Figure 3B). It has been known that platelet aggregation and dense granule secretion only require a positive-feedback effect of generated TxA_2_ when stimulated with a low concentration of thrombin [34,35]. Taken together, our data suggest that prednisolone affects low concentration of thrombin-induced platelet aggregation and dense granule secretion by regulating the positive-feedback effect of TxA_2_ generation.

### 3.4. Prednisolone Inhibits 2-MeSADP- and Thrombin-Induced TxA_2_ Generation in Platelets

To confirm the effect of prednisolone on TxA_2_ generation, we stimulated the platelets with 2-MeSADP and thrombin and measured the amount of TxA_2_ release. As shown in Figure 4, both low and high concentrations of 2-MeSADP- and thrombin-induced TxA_2_ generation were significantly inhibited in the presence of prednisolone, confirming the inhibitory effect of prednisolone on TxA_2_ generation.

### 3.5. Prednisolone Inhibits 2-MeSADP-Induced cPLA_2_ and ERK Phosphorylation

ERK phosphorylation is one of the most important upstream signaling molecules in TxA_2_ generation in platelets [36]. To understand the molecular mechanism involved in the regulation of TxA_2_ generation by prednisolone, we stimulated the platelets with 2-MeSADP and measured the cPLA_2_ ERK phosphorylation in non-aspirinated and aspirinated platelets. As shown in Figure 5, prednisolone completely inhibited 2-MeSADP-induced cPLA_2_ phosphorylation in both non-aspirinated and aspirinated platelets. However, 2-MeSADP-induced ERK phosphorylation was significantly inhibited by prednisolone only in non-aspirinated platelets, while prednisolone had no effect on 2-MeSADP-induced ERK phosphorylation in aspirinated platelets where there is no contribution of generated TxA_2_. These results confirm that prednisolone directly inhibits cPLA2 phosphorylation to regulate TxA_2_ generation in platelets.

## 4. Discussion

Glucocorticoids are one of the most commonly used steroid medications to treat numerous types of allergies, inflammatory diseases, autoimmune disorders, and cancers [37]. Risks associated with glucocorticoid therapy are associated with dose, duration of therapy, and specific therapy used. Furthermore, chronic glucocorticoid administration can result in toxicities and undesirable effects such as Cushing’s syndrome even at physiological doses [38]. Thus, it is important to comprehend the precise mechanisms of the pathophysiological and therapeutic effects of glucocorticoids. One of the various side effects of glucocorticoid administration is hypercoagulopathy, which is known to be triggered by thrombocytosis, hyperglycemia, hypertension, and dyslipidemia [8,9,39]. However, there are discrepancies on how glucocorticoids affect platelet function, which is known to play a crucial role in coagulopathy. In contrast to the known role of glucocorticoids in hypercoagulability, previous studies have reported the inhibitory effect of glucocorticoids on platelet function using PRP [28,29,30,40]. PRP contains various coagulative factors that influence platelet functional responses. To rule out the contribution of the contents in PRP and thoroughly understand the molecular mechanism involved in the regulation of platelet function by glucocorticoids, we used a washed platelet system and investigated the effect of prednisolone, one of the most commonly used glucocorticoids in platelets.

Sex hormones, another type of steroid, have a well-established role in platelet function [41,42,43]. As corticosteroids and sex hormones are generated from cholesterol via the same steroidogenic pathway, they share similar effects on target cells [44,45]. 17β-estradiol and progestin, sex hormones, have been known to inhibit ADP-induced platelet activation [41]. Consistently, we observed that 2-MeSADP-induced platelet dense granule secretion and the resultant secondary wave of aggregation were completely inhibited in the presence of prednisolone. It has been shown that ADP-induced secretion and the resultant secondary wave of aggregation are dependent on the positive-feedback effect of generated TxA_2_ [46]. When the contribution of TxA_2_ generation was blocked by aspirin treatment, prednisolone had no further inhibitory effect on 2-MeSADP-induced platelet aggregation and secretion compared to non-aspirinated platelets, suggesting that prednisolone inhibits the 2-MeSADP-induced secondary wave of aggregation and secretion through the inhibition of the TxA_2_ generation in platelets.

Thrombin is the most potent agonist that activates platelets via the PAR-mediated pathway [47]. It has been shown that 17β-estradiol and progestin inhibit thrombin-induced platelet activation as well [41]. In contrast, others have reported that 17β-estradiol and progesterone potentiate thrombin-induced platelet activation [42,43]. We found that thrombin-induced platelet aggregation and dense granule secretion were inhibited in the presence of prednisolone. However, unlike the effect of prednisolone on 2-MeSADP-induced platelet function, we observed that prednisolone only inhibited low-thrombin concentration-induced platelet aggregation and secretion, and only partially inhibited high-thrombin concentration-induced secretion. In contrast to ADP, it has been known that TxA_2_ generation and the subsequent granule secretion play a role at low-thrombin concentration-induced platelet activation [48]. Therefore, our data suggest that prednisolone affects thrombin-induced platelet function by explicitly inhibiting TxA_2_ generation. More importantly, we found that 2-MeSADP-and thrombin-induced TxA_2_ generation was significantly inhibited by prednisolone, confirming the effect of prednisolone on TxA_2_ generation.

Glucocorticoids are known to inhibit cPLA_2_ in several cells [49,50]. cPLA_2_ is an upstream molecule of TxA_2_ generation that separates arachinonic acid, a precursor of TxA_2_, from phospholipids [51]. Glucocorticoids have also been shown to activate ERK1/2 in PC12 cells (a pheochromocytoma cell) and vascular smooth muscle cells [52,53,54]. However, in mast cells, dexamethasone has suppressed the phosphorylation of proteins that are associated with the activation of the mitogen-activated protein kinases (MAPKs) [55]. So far, the effect of glucocorticoids on cPLA_2_ and MAPKs has not been determined in platelets. In our study, we found that 2-MeSADP-induced platelet cPLA_2_ phosphorylation was completely inhibited by prednisolone in both non-aspirinated and aspirinated platelets. However, 2-MeSADP-induced ERK phosphorylation was significantly inhibited only in non-aspirinated platelets, whereas no additional inhibitory effect of prednisolone on ERK phosphorylation was observed in aspirinated platelets. It has been known that ERK plays a major role in TxA_2_ generation downstream of P2Y_1,_ P2Y_12,_ and PARs in platelets [22,33,56]. Further, it has been known that TxA_2_ can induce ERK phosphorylation by itself in platelets [57]. Thus, our data indicate that prednisolone affects platelet activation by inhibiting TxA_2_ generation by directly regulating cPLA_2_ phosphorylation, and that the effect of prednisolone on ERK phosphorylation is mediated by inhibition of TxA_2_ generation in platelets. High glucocorticoids have been shown to be effective in combination with first-line medication for some bleeding disorders because of their coagulation function [27]. It has also been demonstrated that in a majority of bleeding disorders, it is difficult to control the use of glucocorticoids as it may also lead to hypercoagulability such as in venous thromboembolism [58]. Furthermore, using glucocorticoids has been linked to the development of a number of metabolic disorders [59]. In contrast, glucocorticoids have also been known to cause bleeding disorders in the uterus and gastrointestinal tract [58,60]. Our study has demonstrated that prednisolone inhibits platelet activity, pointing to its potential role in hypo-coagulation and contradicting the prevalent notion that glucocorticoids promote coagulation.

## 5. Conclusions

Cushing’s syndrome is a disease that severely affects both humans and dogs and is potentially life-threatening if it remains untreated [61]. It significantly reduces the quality of life and, thus, requires effective treatment [62]. In this study, we demonstrate that prednisolone, a glucocorticoid, inhibits platelet function by the inhibition of TxA_2_ generation through the regulation of cPLA_2_ phosphorylation. Moreover, it indirectly regulates generated TxA2-mediated ERK phosphorylation to control platelet functional responses. Overall, our study provides evidence that could be applied to the therapeutic use of prednisolone in coagulation and bleeding disorders as well as the clinical management of Cushing’s syndrome in dogs in the future.

## Figures and Tables

**Figure 1 animals-13-01299-f001:**
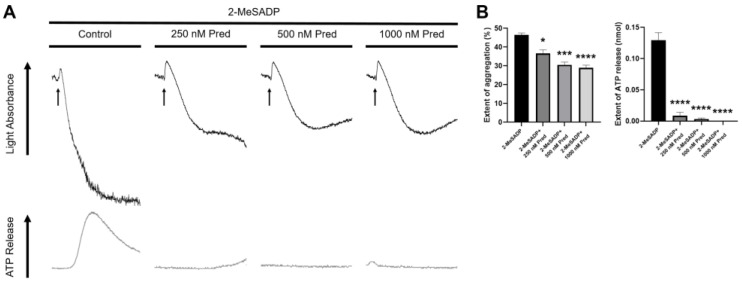
The effect of prednisolone on 2-MeSADP-induced platelet aggregation and secretion. (**A**) Washed murine platelets were stimulated with 100 nM 2-MeSADP in the presence of different concentrations of prednisolone for 3.5 min under stirring conditions. Platelet aggregation and dense granule secretion were analyzed by a Lumi-aggregometer. The entire set of data represents three separate experiments (*n* = 3). (**B**) The extent of aggregation and dense granule secretion were quantified from panel A. Data are shown as mean ± SE. *, *p* < 0.05; ***, *p* < 0.001; ****, *p* < 0.0001.

**Figure 2 animals-13-01299-f002:**
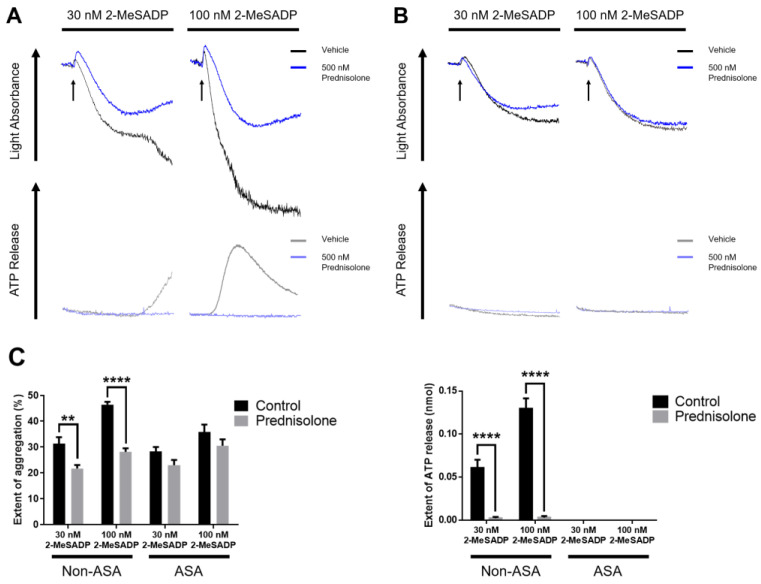
The effect of prednisolone on 2-MeSADP-induced platelet aggregation and secretion in aspirinated platelets. (**A**) Non-aspirinated and (**B**) aspirinated washed platelets were stimulated with 30 nM and 100 nM 2-MeSADP in the presence or absence of 500 nM prednisolone and platelet aggregation and secretion were measured. The entire set of data represents three separate experiments (*n* = 6). (**C**) The extent of aggregation and dense granule secretion were quantified from panels A and B. Data are shown as mean ± SE. **, *p* < 0.01; ****, *p* < 0.0001.

**Figure 3 animals-13-01299-f003:**
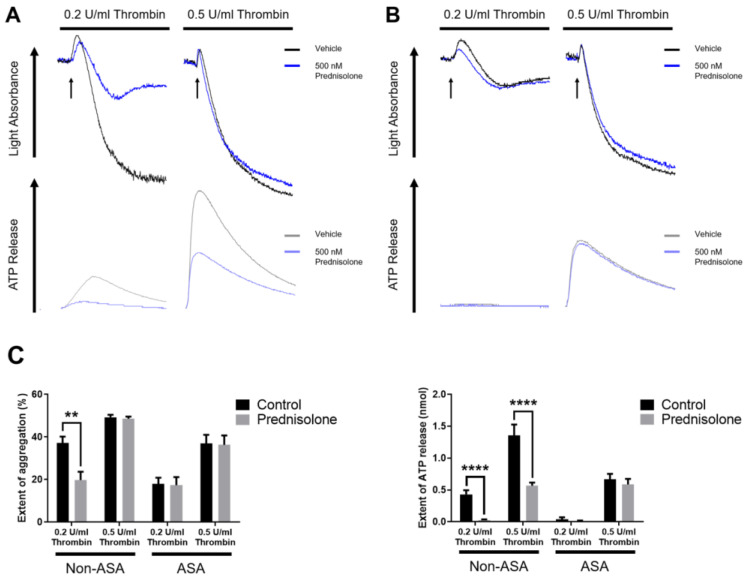
The effect of prednisolone on thrombin-induced platelet aggregation and secretion. (**A**) Non-aspirinated and (**B**) aspirinated washed platelets were stimulated with 0.2 U/mL and 0.5 U/mL thrombin in the presence or absence of 500 nM prednisolone, and platelet aggregation and secretion were measured. The entire set of data represents three separate experiments (*n* = 6). (**C**) The extent of aggregation and dense granule secretion were quantified from panels A and B. Data are presented as mean ± SE. **, *p* < 0.01; ****, *p* < 0.0001.

**Figure 4 animals-13-01299-f004:**
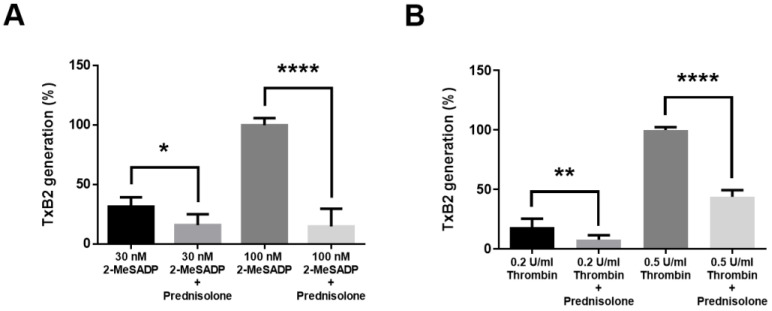
The effect of prednisolone on TxA_2_ generation induced by 2-MeSADP and thrombin in platelets. Washed platelets were stimulated with different concentrations of (**A**) 2-MeSADP and (**B**) thrombin for 3.5 min in the presence or absence of 500 nM prednisolone and TxB_2_ generation was measured (*n* = 6). Data were expressed as percentages compared to 100 nM 2MeSADP and 0.5 U/mL thrombin groups and are shown as mean ± SE. *, *p* < 0.05; **, *p* < 0.01; ****, *p* < 0.0001.

**Figure 5 animals-13-01299-f005:**
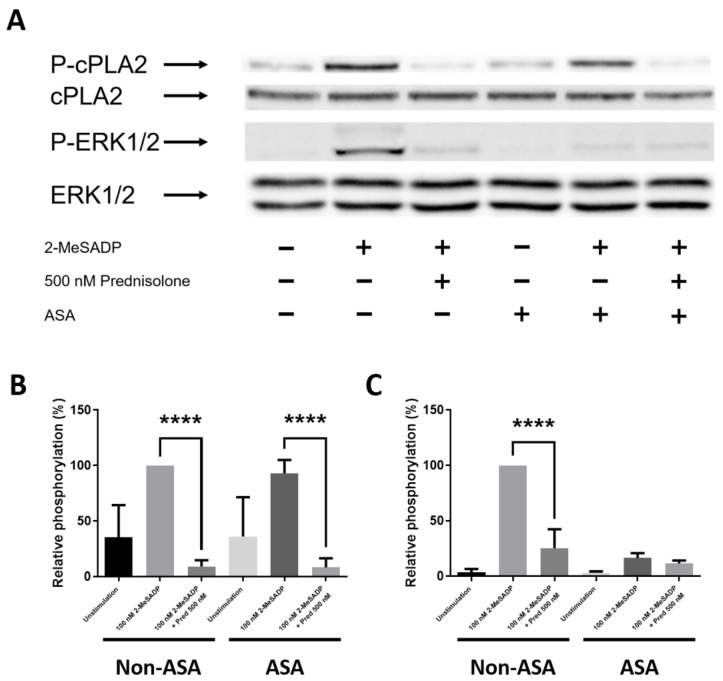
The effect of prednisolone on 2-MeSADP-induced cPLA_2_ and ERK phosphorylation in platelets. (**A**) Non-aspirinated and aspirnated washed platelets were stimulated with 100 nM 2-MeSADP in the presence of 500 nM prednisolone. Proteins were immunoblotted using anti-phospho-cPLA_2_, anti-cPLA_2_, anti-phospho-ERK, and anti-ERK antibodies (*n* = 9). Relative phosphorylation of (**B**) cPLA_2_ and (**C**) ERK were expressed as a percentage compared to the 100 nM 2-MeSADP group in non-aspirinated platelets and are shown as mean ± SE. ****, *p* < 0.0001.

## Data Availability

The data presented in this study are available on request from the corresponding author.

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
