# Peer review of "Role of Prednisolone in Platelet Activation by Inhibiting TxA2 Generation through the Regulation of cPLA2 Phosphorylation"

_animals, 2023, doi:10.3390/ani13081299_

Round 1

Reviewer 1 Report

This study uses washed mouse platelets to study the in vitro effects of the steroid prednisolone on platelet function. They determine that pred impairs platelet function by inhibiting ERK signaling and TxA2 generation, which acts similarly to aspirin and also impacts the platelet response to ADP and low dose thrombin. The study is simple, well designed, and the manuscript presents the data clearly. Several comments:

1) One limitation in the interpretation of the data is that ADP does not directly induce secretion in mouse platelets as it does in human platelets due to lack of PKC activation. Therefore, human platelets may be able to overcome the lack of ERK signaling and TxA2 generation when stimulated with ADP and may have a lesser extent of impairment than mouse platelets.

2) The authors should speculate or provide literature on steroid receptors in platelets- what is the receptor for prednisolone (and corticosteroids) in platelets and how is it linked to ERK signaling?

3) Is there any data on plasma TxB2 (stable TxA2 metabolite) in ITP patients on prednisolone? Platelets are the major source of basal plasma TxB2.

4) The content of methods section 2.5 is a copy of section 2.1 and should be corrected.

Author Response

We would like to thank the reviewer for the evaluation of the manuscript and helpful suggestions. We are thankful to the reviewers for the insightful comments and criticisms. We have clarified the issues raised by reviewers. The responses to the specific comments for the reviewers are given below:

Reviewer 1

This study uses washed mouse platelets to study the in vitro effects of the steroid prednisolone on platelet function. They determine that pred impairs platelet function by inhibiting ERK signaling and TxA2 generation, which acts similarly to aspirin and also impacts the platelet response to ADP and low dose thrombin. The study is simple, well designed, and the manuscript presents the data clearly. Several comments:

Comment 1: One limitation in the interpretation of the data is that ADP does not directly induce secretion in mouse platelets as it does in human platelets due to lack of PKC activation. Therefore, human platelets may be able to overcome the lack of ERK signaling and TxA2 generation when stimulated with ADP and may have a lesser extent of impairment than mouse platelets.

Response: We thank the reviewer for the comment. Platelet activation occurs through a complicated signaling pathway with multiple positive feedback loops in an agonist-dependent manner. It is well known that in both human and mouse platelets, ADP-stimulated platelets require TxA2 generation that causes activation of TP receptor-mediated signaling events through an autocrine loop that leads to ADP secretion. (Jin et al., 2002). In regards to differences in PKC expression and its activation in human and mouse platelets, only PKC δ is relatively high and PKC ε is not expressed in human platelets compared to mouse platelets (Pears et al. 2008). PKC δ isoform is known to regulate ADP-induced TxA2 generation in platelets (Kunapuli et al., 2017). This generated TxA2 further causes dense granule secretion in ADP-stimulated platelets. (Chari et al., 2009). In addition, positive feedback effect of generated TxA2 on ADP-induced secretion and the resultant secondary wave of aggregation is similar in both human and mouse platelets. Since we have shown that prednisolone inhibits TxA2 generation, we believe that the inhibitory effect of prednisolone on ADP-induced platelet functional responses would be similar in both human and mouse platelets.  

Comment 2: The authors should speculate or provide literature on steroid receptors in platelets- what is the receptor for prednisolone (and corticosteroids) in platelets and how is it linked to ERK signaling?

Response: We thank the reviewer for the comment. The receptors for prednisolone and the associated signaling pathway in platelet is not well known. But it has been reported that platelets express glucocorticoid receptors (GRs) in their cytosol and membrane. However, its role in platelet signaling and how it is related to ERK has not been known yet. Initially, we checked 2-MeSADP-induced ERK phosphorylation in the presence of prednisolone and found that prednisolone inhibits 2-MeSADP-induced ERK phosphorylation in non-aspirinated platelets, suggesting that prednisolone may regulate ADP-induced platelet function by regulating ERK phosphorylation. During the revision, we checked the cPLA2 phosphorylation since cPLA2 is the rate-limiting factor for TxA2 generation in platelets and found that prednisolone inhibits cPLA2 phosphorylation in both non-aspirinated and aspirinated platelets. In addition, we found that prednisolone only inhibits 2-MeSADP-induced ERK phosphorylation in non-aspirinated platelets whereas it has no effect on ERK phosphorylation in aspirinated platelets where there is no contribution of generated TxA2. These suggest that prednisolone affects the TxA2 generation by directly targeting the cPLA2 phosphorylation and the inhibition of ERK phosphorylation by prednisolone may be due to the inhibition of the positive-feedback effect of generated TxA2 in platelets. This has been discussed in the revised version of the manuscript.

Comment 3: Is there any data on plasma TxB2 (stable TxA2 metabolite) in ITP patients on prednisolone? Platelets are the major source of basal plasma TxB2.

Response: We thank the reviewer for the comment. Data on plasma TxB2 in ITP patients on prednisolone have not been reported. Glucocorticoids like prednisolone usually increase the absolute number of platelets in ITP patients. However, even if the total number of platelets increases, we expect the decreased plasma TxB2 in ITP patients on prednisolone based on our results showing the inhibitory effect of prednisolone on thromboxane generation.

Comment: 4) The content of methods section 2.5 is a copy of section 2.1 and should be corrected.

Response: We thank the reviewer for the comment. We have corrected the manuscript. Please refer to the revised version of the manuscript.

Reviewer 2 Report

This is an interesting manuscript with potentially valuable information, although it may be better suited in a different journal. Of concern however, is the assumption that prednisolone decreases thromboxane A2 production while disregarding the role of ADP in the process. The ADP analogue (2-MeS ADP) stimulates the 2Y1 receptors leading to the ultimately activation of the GP receptors leading to aggregation. The process of thromboxane production occurs in concert with ADP. It is important to explain whether prednisolone decreases thromboxane production directly, or whether the decreased ADP activity (which has been found previously - Liverani et al. Biochem Pharmacol. 2012;83(10):1364-1373) influences the degree of thromboxane release and production. This can greatly alter the premise for the conclusions made, but can be rewritten and resubmitted for publication.

Some suggestions to improve the overall quality of the manuscript.

Line 9: "Although" might be better suited than "Contrary".

Line 10: Consider an alternative to "molecular basis" - perhaps "mechanism of action".

Line 18: "determined to characterize" - replace with "studied".

Line 19: Remove "molecular basis" from the sentence.

Line 46: Please rewrite the sentence "Among the various side effects of glucocorticoid over-use, thrombotic, hemostatic, and cardiovascular diseases account for an important part due to their high morbidity and mortality rates" - "Excessive use of glucocorticoids in thrombotic, heamostatic and cardiovascular diseases are associated with increased morbidity and mortality".

Line 82: "Till now, the mechanisms involved in the conflicting effects of glucocorticoids on platelet function and the associated molecular mechanism involved are not elucidated and needs further investigation." - It has been studied elsewhere and relatively recently - rather rewrite this sentence to say "the effect on the production of thromboxane has not been elucidated."

Line 105: Please include more detail on how the mice were housed and the conditions and food they had access to. This is crucial information where an anti-inflammatory agent is studied to ensure that no confounding factors may be present in the environment. Were the mice terminated for this study?

Line 145: Please include what statistical tests were used to determine whether the data had a normal spread. Also indicate the confidence interval and the p-value considered significant.

Figure 1: Why was ATP levels determined? Was this meant to be an indication of ADP release? If so, please include this specifically in the Materials and Methods section.

Figure 2: Was a comparison done between the Non-ASA and ASA? And not just the control and the prednisolone - this comparison would also be interesting to determine what the effect of the prednisolone was when thromboxane production was also inhibited? This would be valuable to determine whether prednisolone had an effect on thromboxane production or mainly ADP.

Line 176: It is stated that "It is well established that ADP-induced platelet secretion and the secondary wave of aggregation require TxA2 generation" - this is a very old reference (1968). Here again the focus seems to be on the role of ADP on TXA2 generation and not directly on TXA2 generation. Please use an alternative reference here and later on in the manuscript as well.

Line 248: "Unfortunately, the risk of adverse effects from glucocorticoid therapy is emerged by dose, duration of therapy, and the specific agent used." Please rewrite as "Risks associated with glucocorticoid therapy is associated with dose, duration of therapy and specific therapy used."

Author Response

We would like to thank the reviewer for the evaluation of the manuscript and helpful suggestions. We are thankful to the reviewers for the insightful comments and criticisms. We have clarified the issues raised by reviewers. The responses to the specific comments for the reviewers are given below:

Reviewer 2

Comment: This is an interesting manuscript with potentially valuable information, although it may be better suited in a different journal. Of concern however, is the assumption that prednisolone decreases thromboxane A2 production while disregarding the role of ADP in the process. The ADP analogue (2-MeSADP) stimulates the 2Y1 receptors leading to the ultimately activation of the GP receptors leading to aggregation. The process of thromboxane production occurs in concert with ADP. It is important to explain whether prednisolone decreases thromboxane production directly, or whether the decreased ADP activity (which has been found previously - Liverani et al. Biochem Pharmacol. 2012;83(10):1364-1373) influences the degree of thromboxane release and production. This can greatly alter the premise for the conclusions made, but can be rewritten and resubmitted for publication.

Response: We thank the reviewer for the comment. Our study will shed lights on the pathophysiological and therapeutic effects of glucocorticoids in Cushing's syndrome in dogs. We have added contents regarding the importance of our study in dogs with hypercortisolism that fits within the scope of the journal. According to Liverani et al., prednisolone selectively regulated ADP-induced P2Y12 receptor signaling. If prednisolone had inhibitory effect on ADP-induced signaling, we would have seen the additional inhibition of ADP-induced platelet aggregation in the presence of prednisolone in Figure 2B where there is no contribution of generated TxA2. More importantly, we checked the cPLA2 phosphorylation in the revision and found that prednisolone inhibits cPLA2 phosphorylation in both non-aspirinated and aspirinated platelets confirming that prednisolone affects the TxA2 generation by directly targeting the cPLA2 phosphorylation since cPLA2 is the rate-limiting factor for TxA2 generation in platelets. Additionally, we showed that prednisolone only inhibits 2-MeSADP-induced ERK phosphorylation in non-aspirinated platelets whereas it has no effect on ERK phosphorylation in aspirinated platelets where there is no contribution of TxA2. These suggest that the inhibition of ERK phosphorylation by prednisolone may be due to the inhibition of the positive-feedback effect of generated TxA2 in platelets.

Some suggestions to improve the overall quality of the manuscript.

Comment: Line 9: "Although" might be better suited than "Contrary".

Response: We thank the reviewer for the comment. We have corrected the manuscript. Please refer to the revised version of the manuscript.

Comment: Line 10: Consider an alternative to "molecular basis" - perhaps "mechanism of action".

Response: We thank the reviewer for the comment. We have corrected the manuscript. Please refer to the revised version of the manuscript.

Comment: Line 18: "determined to characterize" - replace with "studied".

Response: We thank the reviewer for the comment. We have corrected the manuscript. Please refer to the revised version of the manuscript.

Comment: Line 19: Remove "molecular basis" from the sentence.

Response: We thank the reviewer for the comment. We have corrected the manuscript. Please refer to the revised version of the manuscript.

Comment: Line 46: Please rewrite the sentence "Among the various side effects of glucocorticoid over-use, thrombotic, hemostatic, and cardiovascular diseases account for an important part due to their high morbidity and mortality rates" - "Excessive use of glucocorticoids in thrombotic, heamostatic and cardiovascular diseases are associated with increased morbidity and mortality".

Response: We thank the reviewer for the comment. We have corrected the manuscript. Please refer to the revised version of the manuscript.

Comment: Line 82: "Till now, the mechanisms involved in the conflicting effects of glucocorticoids on platelet function and the associated molecular mechanism involved are not elucidated and needs further investigation." - It has been studied elsewhere and relatively recently - rather rewrite this sentence to say "the effect on the production of thromboxane has not been elucidated."

Response: We thank the reviewer for the comment. Since the effect of prednisolone on platelet function is contradictory and not clearly elucidated yet, we revised the sentence as “Till now, the mechanisms involved in the effects of glucocorticoids on the production of thromboxane in platelets has not been elucidated.". Please refer to the revised version of the manuscript.

Comment: Line 105: Please include more detail on how the mice were housed and the conditions and food they had access to. This is crucial information where an anti-inflammatory agent is studied to ensure that no confounding factors may be present in the environment. Were the mice terminated for this study?

Response: We thank the reviewer for the comment. We added the breeding environment and condition of mice following the thoughtful reviewer's advice.

Comment: Line 145: Please include what statistical tests were used to determine whether the data had a normal spread. Also indicate the confidence interval and the p-value considered significant.

Response: We thank the reviewer for the comment. We revised the statistical analysis as below:
We performed the normality test with Shapiro-Wilk test before performing student's t-test and one-way ANOVA, and if the data did not follow normality, we performed the Mann-Whitney test.

Comment: Figure 1: Why was ATP levels determined? Was this meant to be an indication of ADP release? If so, please include this specifically in the Materials and Methods section.

Response: We thank the reviewer for the comment. ATP is co-packaged in platelet dense granules with serotonin, Ca2+, and ADP. By luciferin-luciferase assay, we can detect ATP by light formation enzymatically generated by luciferase (ATP + luciferin + O2 + luciferase -> AMP + PPi + CO2 + oxyluciferin + light). As per the reviewer’s suggestion, we have revised the Material and Methods section.

Comment: Figure 2: Was a comparison done between the Non-ASA and ASA? And not just the control and the prednisolone - this comparison would also be interesting to determine what the effect of the prednisolone was when thromboxane production was also inhibited? This would be valuable to determine whether prednisolone had an effect on thromboxane production or mainly ADP.

Response: We thank the reviewer for the comment. Actually, in Figure 2, we compared the 2-MeSADP-induced platelet aggregation and secretion in the presence and absence of prednisolone in Non-ASA- and ASA-treated platelet. We found that 2-MeSADP-induced platelet aggregation and dense granule secretion is inhibited in the presence of prednisolone in non-aspirinated platelets. Further, prednisolone showed no further inhibition of 2-MeSADP-induced platelet aggregation in aspirinated platelets where there is no contribution of generated TxA2, indicating that prednisolone inhibits TxA2 generation to regulate ADP-induced platelet function rather than affecting the ADP-receptor itself. We have also clarified whether prednisolone has an effect on TxA2 production or mainly ADP in the response to the Reviewer’s 1st comment.  

Comment: Line 176: It is stated that "It is well established that ADP-induced platelet secretion and the secondary wave of aggregation require TxA2 generation" - this is a very old reference (1968). Here again the focus seems to be on the role of ADP on TXA2 generation and not directly on TXA2 generation. Please use an alternative reference here and later on in the manuscript as well.

Response: We thank the reviewer for the comment. As we mentioned earlier, we found the direct target of prednisolone, the cPLA2, in the revised version of the manuscript and confirmed the direct inhibitory effect of prednisolone on 2-MeSADP-induced TxA2 generation. We have corrected the reference in the manuscript. Please refer to the revised version of the manuscript.

Comment: Line 248: "Unfortunately, the risk of adverse effects from glucocorticoid therapy is emerged by dose, duration of therapy, and the specific agent used." Please rewrite as "Risks associated with glucocorticoid therapy is associated with dose, duration of therapy and specific therapy used."

Response: We thank the reviewer for the comment. We have corrected the manuscript. Please refer to the revised version of the manuscript.

Round 2

Reviewer 2 Report

I would like to recommend this article for publication in its current form.

Author Response

We are thankful to the reviewer for re-evaluating the manuscript and recommending acceptance.